# Optimization of Vacuum Microwave-Assisted Extraction of Pomegranate Fruits Peels by the Evaluation of Extracts’ Phenolic Content and Antioxidant Activity

**DOI:** 10.3390/foods9111655

**Published:** 2020-11-12

**Authors:** Prodromos Skenderidis, Stefanos Leontopoulos, Konstantinos Petrotos, Ioannis Giavasis

**Affiliations:** 1Department of Agrotechnology, Laboratory of Food and Biosystems Engineering, University of Thessaly, 41110 Larissa, Greece; s_leontopoulos@yahoo.com (S.L.); petrotos@uth.gr (K.P.); 2Department of Food Technology, Laboratory of Food Microbiology and Biotechnology, University of Thessaly, End of N. Temponera Street, 43100 Karditsa, Greece; igiavasis@uth.gr

**Keywords:** pomegranate peels, antioxidants, vacuum microwave extraction, response surface methodology, polyphenols, radical scavenging, valorization

## Abstract

The global interest in the use of plant by-product extracts as functional ingredients is continuously rising due to environmental, financial and health benefits. The latest advances in extraction technology have led to the production of aqueous extracts with high bioactive properties, which do not require the use of organic solvents. The purpose of this study was to optimize the conditions applied for the extraction of pomegranate peels (PP) via a “green” industrial type of vacuum microwave-assisted aqueous extraction (VMAAE), by assessing the potential bioactivity of the extracts (in terms of phenolic content and antioxidant activity), using a response surface methodology. The extraction conditions of temperature, microwave power, time and water/PP ratio were determined by the response surface methodology, in order to yield extracts with optimal total phenolics concentrations (TPC) and high antioxidant activity, based on the IC_50_ value of the scavenging of the 2,2-diphenyl-1-picrylhydrazyl (DPPH^●^) radical. The values of the optimum extraction parameters, such as extraction temperature (61.48 and 79.158 °C), time (10 and 12.17 min), microwave power (3797.24 and 3576.47 W) and ratio of water to raw material (39.92% and 38.2%), were estimated statistically for the two responses (TPC and IC_50_ values), respectively. Under these optimal extraction conditions, PP extracts with high TPC ((5.542 mg Gallic Acid Equivalent (GAE)/g fresh PP))/min and radical scavenging activity (100 mg/L (1.6 L/min)) could be obtained. Our results highlighted that the optimized industrial type of VMAAE could be a promising solution for the valorization of the PP by-products.

## 1. Introduction

The prolongation of life expectancy and the adoption by modern humans of Western urban lifestyles have contributed to the extension of lifespan and the subsequent prevalence of diseases, such as different forms of neoplasms and degenerative diseases, which are partly linked to oxidative stress [1]. At the same time, the need for a more circular economy and the protection of the environment via the reuse οf agro-industrial by-products, coupled with the financial benefits of by-product valorization, have led to a growing interest in the utilization of bioactive (and especially antioxidant) food ingredients derived from plant by-products, using green processes (without the use of chemicals) to manufacture novel, natural food ingredients, nutraceuticals, and even pharmaceuticals [2].

One such example is the pomegranate fruit (*Punica granatum* L.), which is among the oldest fruits known worldwide. Pomegranate fruit is consumed globally, both as a fruit and as a juice product, due to its functional (antioxidant) properties and potential benefits to consumer’s health. During juice production, almost 50% of the pomegranate fruit is wasted as by-product and usually discarded without any valorization [3]. However, some studies have mentioned the use of pomegranate peels (PP) as animal feed. Several studies have shown that PP is the fruit part that conations the highest concentration of bioactive compounds. These compounds include phenolics, like ellagitannins, punicalin and flavonoids, and thus they could be used as natural antioxidants and potential preservatives in foods [4,5,6,7,8].

During food processing and preservation, various oxidative reactions may occur, namely chemical, enzymatic and photochemical reactions, which produce reactive oxygen species (ROS). These promote the formation of oxidative substances, which confer undesirable changes to proteins, lipids and carbohydrates, leading to the impairment of sensory properties and reductions in shelf life [9,10]. To crack this problem, the use of antioxidants in the human diet is one of the most commonly used strategies. Antioxidants are substances that can protect lipids, proteins and other sensitive biomolecules from oxidation, preventing the formation of off-flavors in foods and thus increasing their shelf life [11]. The most common synthetic antioxidants added in foods are butylated hydroxyanisole (BHA), butylated hydroxytoluene (BHT), propyl gallate (PG) and tert-butyl hydroquinone (TBHHQ). However, these additives are unstable and volatile at high temperatures and may have adverse effects on consumers’ health [12]. As a consequence, the replacement of the conventional synthetic antioxidants with novel ones, derived from natural plant materials, without any adverse effects, has been a constant goal of the modern food industry [13].

Previous studies have investigated the possible antioxidant and antimicrobial effects of freeze-dried PP, and have reported a correlation of the antimicrobial activity with their phenolic concentration and their high antioxidant effect, based on the scavenging of DPPH^●^ (2,2-diphenyl-1-picrylhydrazyl) and ABTS^●^+ (2,2′-Azino-bis-(3-ethyl-benzothiazoline-sulphonic acid) radicals [14]. Besides, the application of PP in synthetic growth media has shown a fairly good in vitro antimicrobial effect against *Salmonella typhimurium* and *Camplylobacter jejuni* when applied at a concentration of 10 mg/mL, while for *Escherichia coli, Staphylococcus aureus*, *Listeria monocytogenes* and *Clostridium perfringens*, a concentration of 50 mg/mL was needed [14]. Similar results were presented in the studies of Barathikannan et al. [15] and John et al. [16], which showed that the antimicrobial effect of the PPs depended on their phenolic content, the antioxidant activity and the type of microorganism tested.

With regard to potential extraction methods, many studies have examined the pomegranate peel extract (PPE) using different methods, such as simple stirring [17], pressure-applied extraction [18], enzymatic extraction [19], and recently ultrasound-assisted extraction (UAE) [20]. However, disadvantages and limitations, such as low efficiency, high processing time, high cost and environmental considerations regarding residual by-products, have led to the investigation of alternative extraction methods. At the same time, new technologies, such as pulsed electric fields, high hydrostatic pressure, ohmic heating, ultrasounds, and microwave-assisted extraction, contribute to the minimization of the extraction time and optimization of the yield of bioactive compounds. The vacuum microwave aqueous assisted extraction (VMAAE) method has already been applied to some fruit and vegetable peels, such as mango and tomato peels [21,22].

The VMAAE is based on the creation of high-energy electromagnetic waves under vacuum conditions. These waves can increase ionic water mobility and molecular rotation, and the resulting friction and the heat that is produced can alter the cellular structures in food samples, leading to the rapid migration of bioactive compounds from the solid to the aqueous phase [23]. Furthermore, VMAAE is advantageous to the extraction of (thermo-)sensitive substances, since it can operate without the use of organic solvents, under low pressure and temperature conditions [24]. This can reduce the risk of thermal degradation and oxidation of the active compounds. Additionally, VMAAE can boost the process of mass transfer and facilitate the diffusion of active substances into the (aqueous) solvent, utilizing the accumulated pressure [25]. The thermal degradation and oxidation risk can be reduced using a vacuum (under pressure), which lowers the boiling temperature of the solvent. A comparison between VMAAE and standard microwave-assisted extraction (MAE) in the extraction of vitamin C from guava and green pepper, as well as vitamin E from soybean and tea leaves, has shown increases in extraction yields by 53% and 145% (for vitamin C), and 20% and 60% (for vitamin E), respectively [26].

Therefore, VMAAE is a potentially effective method for extracting phenolics from PP in an industrial scale extractor. The investigation of the roles and interactions of different extraction parameters is essential in order to determine the optimal extraction conditions [27]. Moreover, the efficiency of VMAAE can vary among different plant sources, due to differences in the concentration and type of active phytochemicals [28]. So far, the extraction of aqueous soluble phenolic compounds [29] and ethanol/water soluble flavonoids [30] from pomegranate peels has been examined, and a comparison of MAE with the UAE method has been carried out at a laboratory scale [31].

However, the present study is the first attempt, to our knowledge, to study the efficacy of an industrial-scale VMAAE of pomegranate peels and to optimize the extraction process, not only in terms of yield of bioactive compounds (total phenols) and antioxidant activity (based on DPPH assay), but also in terms of economic efficiency, taking into account the operational cost of the extraction process, which is closely related to the extraction rate of the selected responses (in other words, the industrial extraction rate measured in our experiment directly reflects upon the industrial operational cost of the process) For this purpose, based on an experimental design by Box and Behnken, twenty-nine VMAAE experiments were designed and performed in triplicate. At the same time, the experimental design was modeled by developing appropriate equations that can predict the TPC of the extracts and their antioxidant activity, based on their ability to bind free DPPH^●^ radicals.

## 2. Materials and Methods

### 2.1. Pomegranates Peels

The pomegranate peels (PP) were collected as a by-product of the pomegranate juice process conducted at “Rodones S.A.” (Greece) using the “Wonderful” pomegranate fruit variety. The PP were frozen after juice processing and the frozen peels were milled mechanically, using a commercial mill, and divided into samples of 2 kg. All samples were kept at −20 °C until the extraction process.

### 2.2. Chemicals

Folin Ciocalteu 2N reagent, anhydrous crystal-formed sodium carbonate (PubChem CID:10340), gallic acid (PubChem CID:370), 2,2-diphenyl-1-picrylhydrazyl (DPPH^●^) (PubChem CID:74358), and methanol (PubChem CID:887) were used in this study. All were supplied by Sigma Aldrich (St. Louis, MI, USA).

### 2.3. Extraction Methodology

Each sample (2 kg) was extracted using 20, 50 or 80 L of water, depending on the dilution ratio used, as proposed by the experimental design software. The extraction experiments were conducted in “Pella’s Nature P. Co.” facilities (Edessa, Greece), using an industrial-type MAC-75 multimode microwave extractor (Milestone Inc., Sorisole (BG), Italy). The experimental temperature was set at 40, 60 or 80 °C, the microwave power was set at 2000, 4000 or 6000 W, while the processing time was fixed at 10, 50 or 90 min, respectively, based on experimental design conditions. The extraction vacuum was set at 355 mbar for all samples. Each experiment was performed in triplicate. The obtained extracts were filtered and kept in a freezer at −20 °C until further analysis. The samples were centrifuged at 12,000 rpm for 10 min, and the obtained supernatant was used for further analyses.

### 2.4. Methodology of the Determination of TPC

The TPC was expressed as gallic acid (GA), and it was calculated by using a method described by Skenderidis et al. [32]. Briefly, 1.58 mL distilled water was added in 0.02 mL of each extract. The blank reagent was prepared using 1.6 mL of distilled water. Subsequently, 0.1 mL of Folin Chocalteu reagent was added to the samples and shaken vigorously. Additionally, 0.3 mL of Na_2_CO_3_ solution (20% *w/v*) was added after 5 min, and then the mixture was incubated in the dark for 120 min. Finally, the absorbance of the samples was measured at 765 nm, and the TPC was expressed as mg of gallic acid equivalent (GAE)/g of fresh PP, based on the calibration curve obtained from standard GA solutions (of known concentrations).

### 2.5. Determination of Antioxidant Capacity of the PPE (DPPH^●^ Method)

The method of Brand-Williams et al. [33] for the DPPH^●^ was used for the calculation of the overall antioxidant activity of the pomegranate peel extract (PPE). The DPPH radical assay was chosen because it is one of the most commonly used methods of estimation of antioxidant activity, especially among food scientist, due to its well-known ability to react readily with water-soluble antioxidants, like the ones found in pomegranate extracts. According to the assay, each extract was diluted in purified water and then added with 1 mL of fresh methanol solution of DPPH^●^ (100 μM concentration). These reagents were vigorously mixed and incubated in the dark at 25 °C for 20 min. After that, the absorbance was measured at 517 nm. The pure analyzed extracts (without DPPH solution) and the DPPH^●^ methanol solutions were used as blank and control, respectively.

The percentage of radical-scavenging capacity (% RSC) of the tested extracts was calculated using the equation below (Equation (1)):% DPPH^●^ radical scavenging activity = ((Abs *control − * Abs *sample)/*Abs *control*) *×* 100(1)
where Abs *control* and Abs *sample* are the absorbance values of the control and the tested sample, respectively. The radical-scavenging capacity of the extracts, based on the value of the half-maximum inhibitory concentration (IC_50_), was determined using the plot of the % RSC versus the PPE concentration. The IC_50_ estimates the concentration of the extract required for scavenging 50% of total DPPH radical. Thus, lower IC_50_ values correspond to the higher antioxidant activity of the extract. Finally, in order to maximize the productivity of the extract, concerning its total antioxidant capacity, the extract equivalent volume (EEqV) for achieving the reference IC_50_ value of DPPH^●^ radical scavenging activity, equal to 100 mg L^−1^, was defined and expressed (in L) with the following equation (Equation (2)).
(2)$$\mathrm{EEqV}\;R\;{\mathrm{IC}}_{50}\;{\mathrm{DPPH}}^{●}\;=\;(\frac{Vt\times 100}{{\mathrm{IC}}_{50}\;\mathrm{DPPHt}\;})/W$$
where *Vt* is the total volume in liters (L) of liquid extract at a given extraction time, IC_50_ of DPPH is the half-maximum inhibitory concentration at a given extraction time, and *W* is the weight of the extracted fresh PP, that was always equal to 2 (kg).

### 2.6. Box-Behnken Design (BBD) Experiment

The mathematical modeling of BBD was applied to pick the range-factor of testing points used for the extraction parameters. This statistic methodology is a spherical model, widely used for the improvement of extraction processes due to its proven efficacy [34,35].

Four independent variables were chosen for the BBD model, namely, extraction temperature (*X1* = 40, 60 or 80 °C), extraction time (*X2* = 10, 50 or 90 min), microwave power (*X3* = 2000, 4000 or 6000 W) and the ratio of water solvent to PP (*X4* = 10, 25 or 40 L/kg), as presented in Table 1. These variables are based on the following equation (Equation (3)):*x_i_* = *Χ_i_* − *Χ_o_*/Δ*x_i_*, *X_i_* = 1, 2, 3, 4(3)
*x_i_* and *X_i_* are the dimensionless and actual values of the independent variables, respectively. *I*, *X_o_* is the actual value of the independent variables at the central point, and Δ*Xi* is the phase change of *X_i_*, corresponding to the dimensionless value of variance of the function. TPC and the antioxidant activity (based on the IC_50_ of the free radical (DPPH^●^) scavenging) were selected as the responses, due to their well-known dependency on the extraction method [36,37].

The response variables were adapted to a fourth-order polynomial model equation obtained by the response surface methodology (RSM) (Equation (4)).
*Y* = *β*_0_ + ∑^3^_*i*=1_*β_i_X_i_* + ∑^3^_*i*=1_*β_ii_X_ii_*^2^ + ∑^2^_*i* = 1_∑^3^_*j* = *i* + 1_*β_ii_X_i_X_j_*(4)

The response variables (*Y*) were the TCP and the IC_50_ of the DPPH^●^. *X_i_* and *X_j_* were the independent variables that influence the responses, and *β*_0_, *β_i_*, *β_ii_* and *β*_ij_ were the model’s regression coefficients (intercept, linear, quadratic and interaction).

Consequently, the mathematical program Design-Expert (Version 12) (Stat-Ease, Minneapolis, MN, USA) was used to determine the final 29 combinations of set-points that were implemented in triplicate. These 29 experiments are presented in Table 2. The experiments were randomized to maximize the effects of unexplained variability due to variables in the observed responses. Each variable was categorized into three levels: −1 (low), 0 (intermediate) and +1 (high) (Table 1).

### 2.7. Data Analysis

The TPC extraction rate was estimated, taking into account the operation cost. Thus, a slightly modified equation, given by Qu et al. [38], was used for the estimation of the rate of TPC (RTPC) obtained per fresh weight of PP, taking into account the parameters of the extraction time plus the equipment set up time and the volume of the extracted mixtures, as presented in Equation (5).
(5)$$\mathrm{RTPC}\;(\mathrm{mg}\;\mathrm{GAE}/g\;\mathrm{fresh}\;\mathrm{PP})/\min\;=\;\frac{CtVt}{t+K}$$
where *C_t_* is the equilibrium concentration of TPC in the liquid extract at a given extraction time *t* (mg/L)

*V_t_* is the total volume in liters (L) of liquid extract at a given extraction time, *t* is the extraction time in min and *K* is the constant setup time of the machine, which was estimated to be 15 min during the experiments.

Similarly, the rate of the EEqV of IC_50_ DPPH^●^ scavenging (REEqVR IC_50_ DPPH^●^) was calculated based on Equation (6).
(6)$$\mathrm{REEqVR}\;{\mathrm{IC}}_{50}\;{\mathrm{DPPH}}^{●}\;=\;\frac{\mathrm{EEqV}\;\mathrm{of}\;{\mathrm{IC}}_{50}\;{\mathrm{DPPH}}^{●}}{t+K}$$

### 2.8. Statistical Analysis

The ANOVA was conducted by Design-Expert (Version 12) software, and the multiple regression analysis of the experimental data was performed via the surface response method. For extracted PP, the ANOVA analysis yielded two respective surface response equations, correlating the TPC and EEqV R IC_50_ DPPH^●^ values to the four independent extraction parameters. The same program has also been used to estimate the optimum values of TPC, and the IC_50_ of DPPH^●^ radical scavenging of the PP extracts. Furthermore, the MiniTab (17th version) (Minitab, State College, PA, USA) software was used as the statistical tool to perform Pearson correlation and Spearman Rho tests between the DPPH^●^ and the TPC of the PP extracts.

## 3. Results and Discussion

### 3.1. Predicted Models of Bioactivity Indices by RSM

The results of the 29 extraction experiments with regard to the TPC and DPPH^●^ radical scavenging activity of each PPE are given in Table 2, and are expressed as mean values with ± standard deviation of triplicate measurements.

By applying multiple regression analysis to the experimental data, the response variable and the test variables were correlated using the quadratic Equation (4).

The *p*-value (level of significance) was used as a tool in the BBD analysis to verify the significance of each coefficient. The smaller the *p*-value, the greater the corresponding coefficient was [39]. ANOVA was carried out to determine the productivity of the models and the variables, as shown in Table 3.

### 3.2. Optimization of PP Vacuum Microwave-Assisted Aqueous Extraction

Design-Expert (Version 12) software was used to optimize the two following selected responses. One selected response was the maximum TPC/g PP and the second selected response was the optimum value of EEqVIC_50_ DPPH^●^. Table 4 presents the optimum values of these two dependent variables and the prediction Equations (7) and (8), as proposed by the Design-Expert software.
foods-09-01655-t004_Table 4Table 4The predicted values of selected responses (TPC and EEqVR IC_50_ DPPH^●^) at the optimum conditions for PP extraction and prediction equations for each response derived from the application of multiple regression analysis on experimental data using the Design-Expert software package (Version 12). *X1, X2, X3* and *X4* are the coded parameters for extraction temperature (°C), extraction time (min), microwave power (W) and ratio of PP to water (%).Independent Variables*X1**X2**X3**X4*TPC (mgGAE/g Fresh PP)EEqVR IC_50_ DPPH^●^ (L)77.06010.2103165.02038.590146.442
77.05012.4802240.01039.830
74.730Extraction temperature (°C) *X1*, extraction time (min) *X2*, microwave power (W) *X3*, ratio of PP to water (%) *X4.* The prediction equations for each response that derived from the application of multiple regression analysis on experimental data using the Design-Expert software package (Version 12) are presented below.
TPC (mgGAE/gfw) *=* +79.46 + 5.24 *X1* + 11.32 *X2* − 0.0833 *X3 + 2.34 X4* + 11.22 *X2X3* − 22.30 *X2X4* − 2.37*X3X4* + 18.73 *X1*^2^*−* 4.81*Χ2*^2^ − 5.35 *X3*^2^ − 23.88 *X4*^2^ + 12.13 *X2*^2^*X4* − 29.49 *Χ2Χ4*^2^ − 22.30 *Χ3*^2^*Χ4* + 32.24 *Χ2*^2^*Χ4*^2^(7)
EEqV R IC50 DPPH^●^ (L) *= +*37.45 *+* 4.28 *X1 −* 0.2731 *X2 −* 0.3989 *X3 −* 5.36 *X4 −* 3.66 *X1X2 −* 17.27 *X1X3 +* 9.56*X1X4 +* 11.63 *X2X3 −* 8.97 *X2X4 −* 7.25 *X3X4 +* 8.49 *X1*^2^*−* 7.14 *X2*^2^*−* 0.9221 *X3^2^ +* 10.11 *X4*^2^*+* 1.32 *X1*^2^*X2−* 4.16*X1*^2^*X3 −* 3.24 *X1X2*^2^*+* 4.05 *X2X4*^2^*−* 6.44 *X3*^2^*X4+*6.13 *X3X4*^2^ − 6.74 *X1*^2^*X2*^2^(8)

### 3.3. Correlation of TPC with IC_50_ of DPPH^●^

In the present study, significant negative correlations of −0.813 and −0.743 were observed between the estimated TPC and the IC_50_ values (as expected), using Pearson correlation, as well as using Spearman Rho tests (data not shown). The scatter plot of the correlation of these two values is shown in Figure 1. Previous studies have also demonstrated this negative correlation between the TPC and the antioxidant capacity of PP extracts, based on DPPH radical scavenging [40,41], or on other methods of estimation of antioxidant activity [42,43].

### 3.4. Modeling of the Extraction of PP Based on the Operational Costs

Based on the methodology and Equations (5) and (6) presented in Section 2.7, the estimated results were analyzed with ANOVA, using the reduced quartic mode, and the results are shown in Table 5. Additionally, Table 6 presents the optimum values of these two dependent variables and the prediction Equations (9) and (10), as were estimated by the Design-Expert software. 

Many authors have reported the cost-effective extraction of polyphenols derived from PP using convenient extraction methods. In their study, Negi et al. [44] investigated the antioxidant and antimutagenic activity of PP from the Ganesha variety, extracted with a soxhlet method using different solvents. In their study, an increase in extract yield by 4.8% using water as a solvent was achieved after a 4 h extraction. Furthermore, in other studies, the extraction efficiency, expressed as the TPC of extracts, reached 119 and 82.6 mg GAE/g dry weight under constant stirring for 1 h and 4 h of extraction, respectively [17,45].

Recent studies have explored the use of UAE and MAE techniques that are considered to be energy-efficient technologies. Thus, these technologies are increasingly used in the extraction of natural products as alternatives to traditional extraction techniques. Pan et al. [46] have reported that aqueous UAE led to polyphenol yields of 14.8% and 14.5%, after 6 min and 8 min extractions of PP, using continuous and pulse UAE, respectively. On the other hand, Kaderides et al. [31] comparatively studied the UAE and MAE extraction of PP and concluded that MAE was a more efficient extraction method, which yielded 199.4 mg GAE/g dry pomegranate peel, after 4 min of extraction.

The vacuum microwave-assisted extraction (VMAE) that was used in the present work resulted in a high TPC of 137.97 mg GAE/g, after a 10 min extraction (design point 24, Table 2), probably due to the use of fresh pomegranate peel and not dry PP. Since VMAE was conducted at the industrial scale, the utilization of fresh PP is more interesting from an industrial point of view, in order to avoid the PP drying procedure.

The ANOVA clearly demonstrates that the extraction time (*X*2) was a significant parameter during the VMAE operation, in terms of operational costs. Extraction time in MAE was very short, in contrast to the traditional techniques which typically require several minutes up to a half hour. Apart from the economic benefits, the shortened extraction time can help prevent the potential degradation and oxidation of antioxidants [25]. As indicated in Table 6, a time of 10.04 min was optimal for the extraction of PP in the present study, based on the operational costs. That time is higher than the 4 min of optimum extraction time for PP phenolics reported in Kaderides et al. [31]. However, in that study, a different solvent, a lab type microwave extractor and a different methodology were used. In the study of Xie et al. [47], 15 min was selected as the optimal time of microwave extraction of flavonoids from Cyclocarya paliurus leaves, while Wang et al. [48] applied an extraction time of 2–10 min for the Panax ginseng extraction, using a high-pressure microwave-assisted lab extractor, and observed no significant changes in extract yield after 10 min of operation.

The ratio of water to PP (*X4*) was another important parameter that had to be optimized in the present study. Studies have shown that the use of water as a solvent allows water dispersion into the plant cell matrix more efficiently (than other solvents), resulting in the improvement of heating and thus facilitating the transfer of substances to the solvent at higher mass transfer rates [49,50]. The volume of the solvent must be adequate to ensure that the whole sample is submerged in the solvent during the entire microwave procedure [51]. As shown in Table 6, the optimum ratios of water to PP (to optimize the two responses based on the operation cost) were 39.924 and 38.201, respectively.

The interaction with *X*2*X4* was statistically significant for the two responses, based on operational cost. Furthermore, concerning the response of REEqV R IC_50_ DPPH^●^, the interactions of *X1X2*, *X1X3, X1X4, X2X4* and *X3X4* were estimated as significant, according to ANOVA, as presented in Table 6. In order to gain a better understanding of the results of regression equations, graphical representations of three-dimensional response surfaces and two-dimensional contour plots are depicted in Figure 2 and Figure 3. The 2–3D plots were created by plotting the response against two independent variables, using the z-axis, while retaining the other two variables at their zero levels. These graphs are very useful to visualize the relationship between independent and dependent variables and the interactions between two variables. Different shapes of the contour plots indicate different types of interactions between the variables. A circular contour plot means that the interactions between the corresponding variables are negligible. An elliptical contour suggests that the interactions between the corresponding variables are significant [52].

The parameters of microwave temperature (*X*1) and power (*X*3) are interrelated because high microwave power can increase the extraction temperature and increase extraction yield [53]. The extraction efficiency (yield of extract) increased at high temperatures, due to the reduced viscosity and surface tension of the solvent, which facilitates solvent solubilization and penetration (or wetting) of the extracted materials [54,55]. Furthermore, when VMAE was performed in a closed vessel system, like the industrial system of the MAC-75 microwave extractor, the temperature of the boiling point decreased as a result of the vacuum, resulting in a better extraction efficiency, thanks to the improved desorption of solutes from the extracted matrix [51]. In the present study, the extraction performance increased with the rise in temperature until reaching an optimum temperature, above which any further increase in temperature is detrimental to the extraction efficiency (Figure 2a,b). The optimum temperature for the two studied responses (TPC and IC_50_ values of DPPH radical scavenging), based on operation cost, ranged from 61.48 °C for RTPC to 79.158 for the REEqV R IC_50_ DPPH^●^, while different optimum microwave power values were also estimated for the two responses (Table 6).

Apparently, temperatures above 61.48 °C may affect the stability of the polyphenolic compounds [56]. Furthermore, as shown in Table 6, in order to estimate the conditions for optimal REEqV R IC_50_ DPPH^●^ values, a combination of parameters, such as temperature, microwave power and an additional 2 min extraction time, is required (79.158 °C and 3576.47 W), compared with the optimal conditions for RTPC. This may pose a dilemma in choosing between the optimal operating condition for maximum polyphenol concentration, or the optimal conditions for maximum antioxidant effect. The former criterion (optimization of TPC) seems to also be more energy- and cost-efficient, as it allows the use of a lower extraction temperature and extraction time.

**Table 6 foods-09-01655-t006:** The predicted response values at the optimum conditions and the prediction equation for the rate of TPC of PPE/g fresh PP and REEqVR IC_50_ DPPH^●^ responses, derived from the application of multiple regression analysis to experimental data, using the Design-Expert software package (Version 12). *X1*, *X2*, *X3* and *X4* are the coded parameters for extraction temperature (Co), extraction time (min), microwave power (W), and ratio of PP to water.

Independent Variables					
*X1*	*X2*	*X3*	*X4*	RTPC (mgGAE/g Fresh PP)/min	REEqV R IC_50_ DPPH^●^ (L/min)
61.480	10.037	3797.240	39.924	5.542	
79.158	12.127	3576.470	38.201		1.813

Extraction temperature (°C) *X1,* extraction time (min) *X2*, microwave power (W) *X3*, ratio of PP to water (%) *X4.* The prediction equations for each response that derived from the application of multiple regression analysis on experimental data using the Design-Expert software package (Version 12) are presented below.

RTPC = + 1.21 + 0.1024 *X1* − 1.72 *X2* − 0.0391 *X3* + 0.0361 *X4* − 0.1021 *X1X2* + 0.0477 *X1X3* + 0.2283 *X2X3* −0.7728 *X2X4* − 0.0365 *X3X4* − 0.0994 *X1*^2^ + 1.13 *X2*^2^ − 0.3961 *X3*^2^ + 0.4491 *X1*^2^*X2* + 0.0243 *X1*^2^*X3* − 0.0026 *X1X3*^2^− 0.0513 *X2*^2^*X3* + 0.6621 *X2*^2^*X4* + 1.06 *X2X3*^2^− 0.3431 *X3*^2^*X4* + 0.7116 *X1*^2^*X3*^2^ − 0.5249 *X2*^2^*X3*^2^(9)

REEqVR IC_50_ DPPH^●^ = + 0.5762 + 0.0659 *X1* − 0.5227 *X2* − 0.0701 *X3* − 0.0028 *X4* − 0.1064 *X1X2* − 0.2656 *X1X3*+ 0.1471 *X1X4* + 0.2941 *X2X3* − 0.1362 *X2X4* − 0.1116 *X3X4* + 0.1306 *X1*^2^ + 0.2125 *X2*^2^ − 0.0138 *X3*^2^ + 0.1552 *X4*^2^+ 0.0601 *X1*^2^*X2* − 0.0753 *X1*^2^*X4* + 0.0157 *X1X2*^2^ − 0.1170 *X2*^2^*X3* + 0.0685 *X2X3*^2^ + 0.0000 *X2X4*^2^ − 0.1786 *X3*^2^*X4*+ 0.1584 *X3X4*^2^ − 0.1414 *X1*^2^*X2*^2^ − 0.0437 *X2*^2^*X3*^2^(10)

The interactions of the *X1* and *X3* extraction parameters and the interactions between the other two parameters were found to be significant for REEqVR IC_50_ DPPH^●^, based on operational costs, as presented in Table 5 and Figure 3. Additionally, significant interactions seem to exist between the parameters of *X2X4*, *X2*^2^*X4*, *X2X3*^2^ and *X1*^2^*X3*^2^, based on the ANOVA analysis of RTPC.

As shown in Figure 2a, the extraction rate of TPC of PP continued to increase with the increase in temperature, and reached a peak at 61.48 °C in the first ten minutes of extraction. This probably occurred because of the high power efficiency of the vacuum of the MAC-75 industrial microwave extractor. When the optimum temperature was exceeded, the RTPC decreased. In spite of the RTPC’s decrease at an elevated temperature (above 61.48 °C), the optimum temperature for the REEqVR IC_50_ DPPH^●^ was close to the highest examined temperature.

Additionally, increasing the microwave extraction power (*X3*), the RTPC and the REEqVR IC_50_ DPPH^●^ increased continuously, reaching an optimum value of 3797.24 and 3576.47 W, respectively, above which any further increase in power was not beneficial (Figure 2b and Figure 3b). This observation can be explained by the fact that the responses have optimum values. Above these values, the applied high microwave power might alter the structure of the extracted material and disturb its molecular interactions with the solvent [57].

## 4. Conclusions

In this study, PP extraction was successfully optimized for industrial-scale operation, using a VMAE method in a microwave industrial extractor. The optimum extraction parameters were estimated based on operation cost, in order to produce extracts of high TPC and high antioxidant activity, based on the IC_50_ of DPPH scavenging radical. Extraction temperature (61.48 and 79.158 °C) and extraction time (10 and 12.17 min) were estimated for the optimal extraction of PP, based on optimal total polyphenol content (TPC) and optimal antioxidant activity, respectively. Apparently, the optimal conditions for these two quality parameters of PP extracts may be somewhat different. Furthermore, optimal microwave power (3797.24 and 3576.47 W) and optimal water ratio to raw material (39.92 and 38.2 L/kg) were estimated for the above two quality parameters, respectively. Under these optimal extraction conditions, we obtained high rates of PP extraction with high TPC values (5.542 mgGAE/g fresh PP/min) and high antioxidant capacity (1.81 L/min of extract equivalent volume at reference IC_50_ of DPPH^●^ radical scavenging activity equal to 100 mg/L)_._

Additionally, the results of the statistical correlation between the TPC and the antioxidant capacity of PPE revealed that there are statistically significant correlations of −0.813 and −0.743 between the two responses, based on Pearson correlation and Spearman Rho tests, respectively. The application of verification tests regarding the optimum extraction conditions was based on the triplicate experiments of the industrial scale, and the high R^2^ of the models can assure not only the high accuracy of the models, but also the fact that the optimized process is readily applicable at the industrial scale, which is one of the most important advantages of the present work.

These results are a valuable tool for the pomegranate juice industry, in order to implement at an industrial scale an efficient and economically viable green extraction method with reduced process time and energy or solvent costs, and maximum quality (as expressed by the TPC or antioxidant capacity), while avoiding the use of harmful solvents for the conversion of a fruit by-product into a functional ingredient with high antioxidant activity.

## Figures and Tables

**Figure 1 foods-09-01655-f001:**
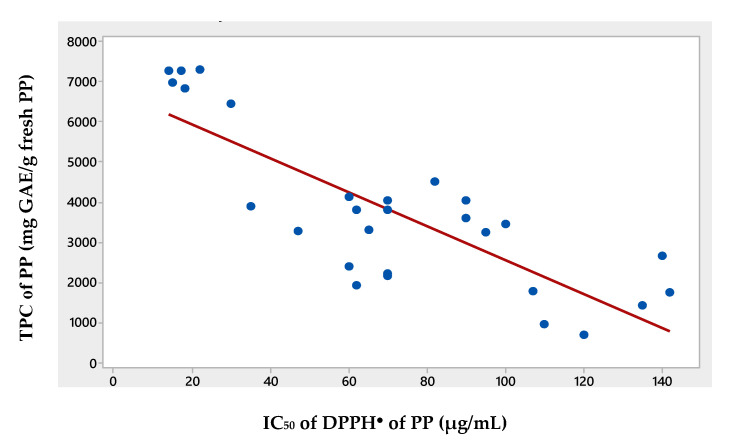
Scatter plot graph of the TPC of PPE and IC_50_ of DPPH^●^ radical scavenging.

**Figure 2 foods-09-01655-f002:**
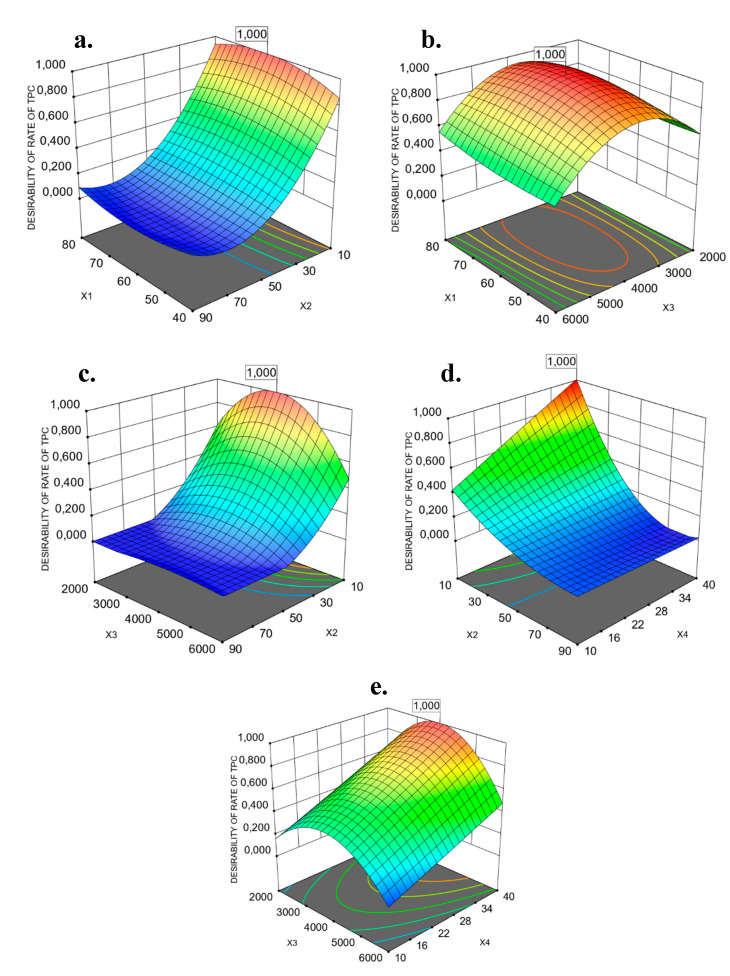
Response surface (3D) with contour plots (2D) showing the interactions of extraction parameters, based on operational costs for RTPC of PPE/g fresh PP. (**a**) Extraction temperature (*X1*) and extraction time (*X2*). (**b**) Extraction temperature (*X1*) and microwave power (*X3*). (**c**) Extraction time (*X2*) and microwave power (*X3*). (**d**) Extraction time (*X2*) and water to solid ratio (*X4*) and (**e**) microwave power (*X3*) and water to solid ratio (*X4*).

**Figure 3 foods-09-01655-f003:**
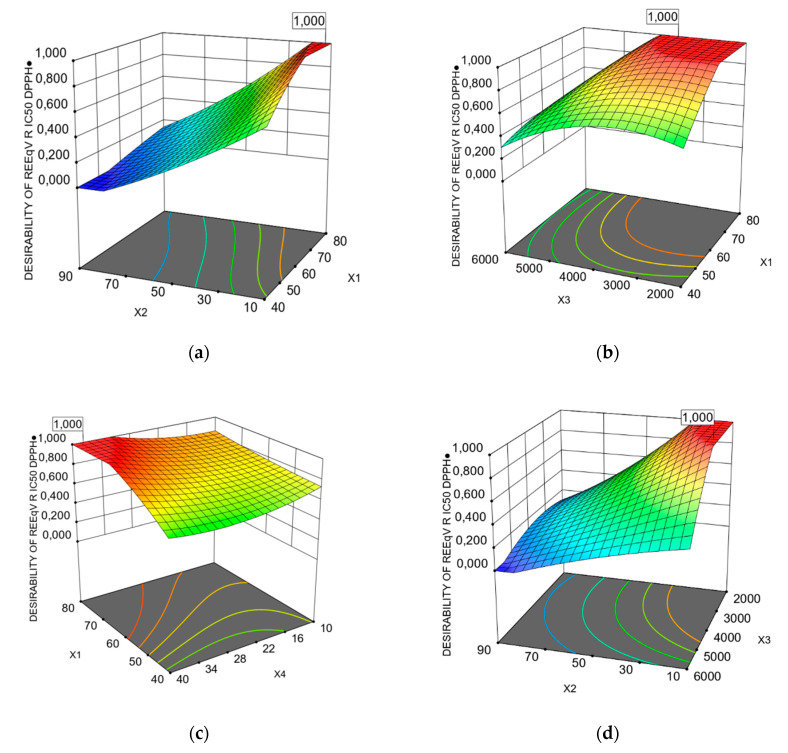
Response surface (3D) with contour plots (2D), showing the interactions of the most significant extraction parameters, based on operational costs for the rate of REEqVR IC_50_ DPPH^●^. (**a**) Extraction temperature (*X1*) and extraction time (*X2*). (**b**) Extraction temperature (*X1*) and microwave power (*X3*). (**c**) Extraction temperature (*X1*) and water to solid ratio(*X4*). (**d**) Extraction time (*X*2) and microwave power (*X*3). (**e**) Extraction time (*X2*) and water to solid ratio(*X4*), (**f**) microwave power (*X3*) and water to solid ratio(*X4*).

**Table 1 foods-09-01655-t001:** Experimental values and coded levels of the independent variables.

Independent Variables	Code Units	Coded Variable Level		
		−1	0	1
Extraction temperature (°C)	*X1*	40	60	80
Extraction time (min)	*X2*	10	50	90
Microwave power (W)	*X3*	2000	4000	6000
Ratio of PP to water (%)	*X4*	10	25	40

**Table 2 foods-09-01655-t002:** Experimental and actual values of the independent variables.

Design Point	Independent Variables in Coded Values				Responses			
					TPC (mgGAE/g Fresh PP)		EEqV R IC_50_ DPPH^●^ (L)	
	*X1*	*X2*	*X3*	*X4*	Measured	Predicted	Measured	Predicted
1	−1	0	1	0	82.29 ± 0.52	74.13	53.19 ± 0.17	53.44
2	1	0	0	1	77.00 ± 0.31	77.00	64.52 ± 1.81	64.54
3	0	0	0	0	55.83 ± 0.08	67.45	35.71 ± 1.46	37.45
4	0	0	−1	1	38.00 ± 0.01	38.00	36.36 ± 0.44	36.36
5	0	0	0	0	82.92 ± 0.28	67.45	38.46 ± 1.12	37.45
6	−1	0	0	1	70.67 ± 0.20	70.67	37.38 ± 0.22	36.86
7	0	0	0	0	54.06 ± 0.61	67.45	35.71 ± 1.46	37.45
8	0	0	1	1	28.17 ± 0.41	28.17	33.33 ± 0.11	33.33
9	0	1	0	1	57.00 ± 0.38	60.49	29.63 ± 0.22	29.88
10	−1	1	0	0	95.52 ± 0.54	89.81	35.71 ± 0.46	35.71
11	1	0	1	0	101.46 ± 0.68	82.87	27.78 ± 1.00	27.48
12	0	−1	0	−1	64.42 ± 0.08	60.16	33.33 ± 1.07	33.03
13	−1	0	−1	0	90.42 ± 0.84	85.79	27.78 ± 0.09	28.03
14	0	0	0	0	101.25 ± 0.84	67.45	35.71 ± 0.25	37.45
15	1	−1	0	0	95.31 ± 0.57	130.31	35.71 ± 1.11	35.71
16	0	−1	−1	0	59.90 ± 0.87	60.32	41.67 ± 0.01	41.69
17	0	1	0	−1	72.67 ± 0.03	80.84	58.82 ± 1.35	58.52
18	0	1	−1	0	66.56 ± 0.37	67.17	17.86 ± 0.37	17.88
19	1	1	0	0	113.02 ± 0.46	105.92	30.49 ± 0.01	30.49
20	−1	−1	0	0	80.94 ± 0.95	104.58	26.32 ± 1.46	26.32
21	0	1	1	0	95.52 ± 0.64	119.07	40.32 ± 0.66	40.35
22	1	0	0	−1	68.50 ± 0.13	68.50	55.56 ± 1.12	56.13
23	1	0	−1	0	97.19 ± 0.98	99.74	71.43 ± 1.41	71.13
24	0	−1	0	1	137.97 ± 0.99	108.87	40.00 ± 0.60	40.25
25	0	−1	1	0	43.96 ± 0.97	45.33	17.61 ± 0.59	17.63
26	0	0	0	0	103.23 ± 0.08	67.45	41.67 ± 1.95	37.45
27	0	0	−1	−1	73.17 ± 0.27	73.17	45.45 ± 0.21	45.45
28	−1	0	0	−1	69.79 ± 0.03	69.79	66.67 ± 0.05	66.69
29	0	0	1	−1	72.83 ± 0.09	72.83	71.43 ± 0.27	71.43

Measured values are medians of three repetitions ± standard deviation. *X1*; Extraction temperature (°C), *X2*: Extraction time (min), *X3*: Microwave power (W), *X4*: Ratio of PP to water (%).

**Table 3 foods-09-01655-t003:** Analysis of variance (ANOVA) for predictive models of TPC (mg GAE/g fresh PP) and EEqV R IC_50_ radical scavenging of PP.

TPC (mg GAE/g Fresh PP)		EEqV R IC_50_ DPPH^●^ (L)	
	*p* Value of the Model		*p* Value of the Model
**Model**	<0.0066 *	**Model**	<0.0001 *
**Variables**	***p* Value**	**Variables**	***p* Value**
*X1*	0.2296	*X1*	0.0006 *
*X2*	0.0445 *	*X2*	0.7964
*X3*	0.9843	*X3*	0.7071
*X4*	0.7498	*X4*	0.0001 *
*Χ2Χ3*	0.1428	*X1X2*	0.0089 *
*X2X4*	0.0085 *	*X1X3*	<0.0001 *
*X3X4*	0.7466	*X1X4*	<0.0001 *
*X1* ^2^	0.0083 *	*X2X3*	<0.0001 *
*X2* ^2^	0.5051	*X2X4*	<0.0001 *
*X3* ^3^	0.3903	*X3X4*	0.0002 *
*X4* ^2^	0.0047 *	*X1* ^2^	<0.0001 *
*X2* ^2^ *X4*	0.2547	*X2* ^2^	0.0002 *
*X2X4* ^2^	0.0053 *	*X3* ^2^	0.3153
*X3* ^2^ *X4*	0.0472 *	*X4* ^2^	<0.0001 *
*X2* ^2^ *X4* ^2^	0.0226 *	*X1* ^2^ *X2*	0.3914
		*X1* ^2^ *X3*	0.0234 *
		*X1X2* ^2^	0.0356 *
		*X2X4* ^2^	0.0260 *
		*X3* ^2^ *X4*	0.0013 *
		*X3X4* ^2^	0.0038 *
		*X1* ^2^ *X2* ^2^	0.0066*
Lack of fitting	0.9987 Not significant	0.9788 Not significant	
*R* ^2^	0.8286	0.9951	
Adj. *R*^2^	0.6309	0.9805	

** p* < 0.05 was considered to be statistically significant. *X1*: Extraction temperature (°C), *X2*: Extraction time (min), *X3*: Microwave power (W), *X4*: Ratio of PP to water (%).

**Table 5 foods-09-01655-t005:** Analysis of variance (ANOVA) of predictive models, concerning PP, TPC, and total equivalent antioxidant volume productivities.

RTPC (mg GAE/g Fresh PP)/min			REEqVR IC_50_ DPPH^●^(L/min)
	*p* Value of the Model		*p* Value of the Model
**Model**	0.0003 * Significant	**Model**	<0.0001 * Significant
**Variables**	***p* Value**	**Variables**	***p* Value**
*X1*	0.3535	*X1*	0.0038 *
*X2*	<0.0001 *	*X2*	<0.0001 *
*X3*	0.7962	*X3*	0.0123 *
*X4*	0.8117	*X4*	0.8830
*X1X2*	0.5063	*X1X2*	0.0021 *
*X1X3*	0.7532	*X1X3*	<0.0001 *
*X2X3*	0.1612	*X1X4*	0.0005 *
*X2X4*	0.0011 *	*X2X3*	<0.0001 *
*X3X4*	0.8093	*X2X4*	0.0007 *
*X1* ^2^	0.5063	*X3X4*	0.0017 *
*X2* ^2^	0.0001 *	*X1* ^2^	0.0008 *
*X3* ^2^	0.0709	*X2* ^2^	0.0004 *
*X1* ^2^ *X2*	0.0657	*X3* ^2^	0.4749
*X1* ^2^ *X3*	0.9095	*X4* ^2^	0.0003 *
*X1X3* ^2^	0.9886	*X1* ^2^ *X2*	0.0686
*X2* ^2^ *X3*	0.8105	*X1* ^2^ *X4*	0.0337 *
*X2* ^2^ *X4*	0.0148 *	*X2* ^2^ *X3*	0.5170
*X2X3* ^2^	0.0013 *	*X2X3* ^2^	0.0064 *
*X3* ^2^ *X4*	0.1399	*X2X4* ^2^	0.0459 *
*X1* ^2^ *X3* ^2^	0.0249 *	*X3* ^2^ *X4*	-
*X2* ^2^ *X3* ^2^	0.0741	*X3X4* ^2^	0.0010 *
		*X1* ^2^ *X2* ^2^	0.0017 *
		*X2* ^2^ *X3* ^2^	0.0120 *
Lack of fitting	0.9228 Not significant		0.7813 Not significant
*R* ^2^	0.9823		0.9985
Adj. *R*^2^	0.9293		0.9915

* *p* < 0.05 was considered to be statistically significant. *X1*: Extraction temperature (°C), *X2*: Extraction time (min), *X3*: Microwave power (W), *X4*: Ratio of PP to water (%).

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
