# Peer review of "Optimization of Vacuum Microwave-Assisted Extraction of Pomegranate Fruits Peels by the Evaluation of Extracts’ Phenolic Content and Antioxidant Activity"

_foods, 2020, doi:10.3390/foods9111655_

Round 1
Reviewer 1 Report
Present study is very interesting, especially due to experiment were conducted in industrial scale and obtained results have applicable value. In spite of the manuscript is pretty good written some improvement of it should be done. Main concern is that whole experiment is based on only two responses variables (authors could give some explanation for that) and also, in manuscript are not shown results of repeated experiment under the optimum conditions to check and compare those results with predicted ones.
Detailed comments:
Title could be rewritten if the main purpose of extraction was extraction of phenolics. Suggestion is: Optimization of vacuum microwave assisted extraction of phenolics from pomegranate fruits peels. According to my opinion is better not to put „bioactivity“ in the title, due to it was determined only by one method. Also, authors in the manuscript could give short explanation why they selected only DPPH – method?
Lines 58, 59 – abbreviation HT, where is full name? Or authors mean on BHT. Is the reference 10 right source of that statement?
Line 79 - Microwave Assisted Extraction
Line 106 – 115 – The purpose of this study should be write more concise. Some information is too detailed and more appropriate to section 2.3. Extraction methodology e.g. “For this reason and based on an experimental design by Box Behnken, twenty-nine VMAAE experiments were designed and performed in triplicate for a pomegranate variety.” The first and the third sentence in that part are very similar, they could be merged.
Lines 151, 152 –The experiments were conducted three times on two separate occasions. What does it mean? And why is written in section 2.5 and not in 2.4.? Suggestion is to write it clearer and put in brackets total number of repetitions for one experiment (sample) and analysis (determinations) (n=…).
Line 166 – when abbreviation is mentioned for the first time in text it should be written in brackets after full name
Calculation for operational cost should be describe in section Methods, too, like 2.7., then: 2.8. ANOVA statistical analysis and estimation of predicted models’ equations. And maybe better title would be “Data analysis”
Table 2 – Are values in column “Measured” average values of triplicate experiment and duplicate determinations? If yes, standard deviations should be written and in fuss note of table should be written something like (if I understand well): Mean ± standard deviation of duplicate determinations from three experiments.
Table 5 and Table 7 – Why “Independent variables” are written above TPC (mgGAE/g fresh PP) and EEqVR IC50 DPPHâ—Ź (L) as well as RTPC (mgGAE/g fresh PP)/min and REEqV R IC50 DPPHâ—Ź (L/min)
Table 5 – Did authors confirmed predicted values experimentally under optimum conditions for PPE? Other words, did authors checked calculated optimum conditions for PPE?
Figure 1 – suggestion is to write units on the graph, and the title inside graph is not necessary.
Lines 241 – 243 – That descriptions should be moved in section Methods due to it presents description of data analysis.
In spite of authors stated similar correlation like e.g. Derakhshan et al. [39], in that study positive correlation was stated. It would be desirable to give better explanation of the negative correlation and of the results in table 2 which are very diverse EEqV R IC50 DPPHâ—Ź (L) and not always accordant to TPC.
Lines 244 – 245 – Did Derakhshan et al. [39], Mansour et 244 al. [40] and Ozgen et al. [41] use the same methods for TPC and the antioxidant activity in PPE like in present study? If no, it should be pointed out.
Line 250 – 266: 3.4 Modelling of the extraction of PP based on the operational costs. Mostly all that part should be moved in section Methods, as I already wrote.
Line 263 and in Eq.6 – EEV? or EEqV
Lines 294 – 296 – Kaderides et al. [29], Xie et al. [46] and Wang et al. [47] did not use the same technique and the same raw material like in present study, so results could not be just compared without pointed out those differences.
Line 321 – three question marks are written, why?
Author Response
Reviewer 1
Comment /Suggestion
Present study is very interesting, especially due to experiment were conducted in industrial scale and obtained results have applicable value. In spite of the manuscript is pretty good written some improvement of it should be done. Main concern is that whole experiment is based on only two responses variables (authors could give some explanation for that) and also, in manuscript are not shown results of repeated experiment under the optimum conditions to check and compare those results with predicted ones.
Αnswer: The two examined responses are linked with our research objectives that have to do with the production of natural antimicrobial and antioxidants for the food industry in industrial type extractor taking account also the extraction’s costs. The TPC is strongly linked with antimicrobial activity while at the same time in many cases the antioxidant activity is linked with TPC (as presented also in our paper), but it may also present some differentiations. For this reason, there are also differentiations between the optimum conditions for the two responses.
Comment /Suggestion
The authors didn’t performed verification test from the extracts received using the optimum conditions but the triplicate experiments and the high R2 of the models can assure the high accuracy of the models.
Αnswer: This is a good suggestion. However, the Box and Behnken Design has a tool that evaluates the effectiveness of the model prediction. This is the repetitions at the central point and the closeness of the obtained values of the responses among the repeated measurements along with the proximity between the Predicted R2 and the adjusted R2 produced by ANOVA analysis (the two values have to have a difference less than 0.2) and the high value of R2 close to unit are according to RSM theory enough proofs for the soundness of the model. But in future work we will take into account the reviewers suggestion and put another proof, using additional measurements at the predicted optimum.
Detailed comments:
Comment /Suggestion
Title could be rewritten if the main purpose of extraction was extraction of phenolics. Suggestion is: Optimization of vacuum microwave assisted extraction of phenolics from pomegranate fruits peels. According to my opinion is better not to put “bioactivity“ in the title, due to it was determined only by one method. Also, authors in the manuscript could give short explanation why they selected only DPPH – method?
Αnswer: The suggestion of the reviewer is correct, so we have changed the title accordingly in the revised manuscript. We chose the DPPH radical because it is most commonly used by other researchers, especially in the food sector. It is also well known that DPPH is reacting better with water soluble antioxidants, which is the case for the pomegranate antioxidants.
«The choice of the word "bioactivity" in the title was based on the fact that in our research we investigated the extraction conditions under which we would receive extracts with a high content of polyphenols in order to use them as natural antimicrobial preservatives for food, while simultaneously we investigated also the extraction conditions in order to receive high antioxidant activity extracts to use it as natural antioxidants. For this reason, it was chosen not to perform the optimization of the extraction conditions for both parameters at the same time. Following reviewer suggestion, we changed the title as Optimization of vacuum microwave-assisted extraction of pomegranate fruits peels and evaluation of extracts’ phenolic content and antioxidant activity.”
Comment /Suggestion
Lines 58, 59 – abbreviation HT, where is full name? Or authors mean on BHT. Is the reference 10 right source of that statement?
Answer: «Τhe abbreviation was corrected as BHT and the reference 10 is the right source of this statement, as reported in the section of ‘Plants as source of antioxidants “ of this review paper “https://www.ncbi.nlm.nih.gov/pmc/articles/PMC3249911/”
Comment /Suggestion
Line 79 - Microwave Assisted Extraction
Answer: Corrected as “Assisted”
Comment /Suggestion
Line 106 – 115 – The purpose of this study should be write more concise. Some information is too detailed and more appropriate to section 2.3. Extraction methodology e.g. “For this reason and based on an experimental design by Box Behnken, twenty-nine VMAAE experiments were designed and performed in triplicate for a pomegranate variety.” The first and the third sentence in that part are very similar, they could be merged.
Answer: Lines 106-115 were rewriten more concisely, according to reviewer remarks.
Comment /Suggestion
Lines 151, 152 –The experiments were conducted three times on two separate occasions. What does it mean? And why is written in section 2.5 and not in 2.4.? Suggestion is to write it clearer and put in brackets total number of repetitions for one experiment (sample) and analysis (determinations) (n=…).
Answer: We deleted it from 2.5 and explained it better, according to the reviewer's suggestion. We mean that each of the 29 extraction experiments was performed 3 times and the mean of the three values of each experiment responses was used in the Design expert statistic software.
Comment /Suggestion
Line 166 – when abbreviation is mentioned for the first time in text it should be written in brackets after full name
Answer: The abbreviation of “BBD” was explained in the title of 2.6
Comment /Suggestion
Calculation for operational cost should be describe in section Methods, too, like 2.7., then: 2.8. ANOVA statistical analysis and estimation of predicted models’ equations. And maybe better title would be “Data analysis”
Answer: We made changes following reviewer’s proposal. The calculations for operational cost were placed in section Methods as 2.7. Data analysis
Comment /Suggestion
Table 2 – Are values in column “Measured” average values of triplicate experiment and duplicate determinations? If yes, standard deviations should be written and in fuss note of table should be written something like (if I understand well): Mean ± standard deviation of duplicate determinations from three experiments.
Answer: Standard deviation was added in column “Measured” of table 2.
Comment /Suggestion
Table 5 and Table 7 – Why “Independent variables” are written above TPC (mgGAE/g fresh PP) and EEqVR IC50 DPPHâ—Ź (L) as well as RTPC (mgGAE/g fresh PP)/min and REEqV R IC50 DPPHâ—Ź (L/min)
Answer: It is now corrected. Τhe alignment of the words “Independent variables” was wrong.
Comment /Suggestion
Table 5 – Did authors confirm predicted values experimentally under optimum conditions for PPE? Other words, did authors checked calculated optimum conditions for PPE?
Answer: This is a good suggestion. However, the Box and Behnken Design has a tool that evaluates the effectiveness of the model prediction. This is the repetitions at the central point and the closeness of the obtained values of the responses among the repeated measurements along with the proximity between the Predicted R2 and the adjusted R2 produced by ANOVA analysis (the two values have to have a difference less than 0.2.) and the high value of R2 close to unit are according to RSM theory enough proofs for the soundness of the model. But in future work we will take into account the reviewers suggestion and put another proof, using additional measurements at the predicted optimum.
Comment /Suggestion
Figure 1 – suggestion is to write units on the graph, and the title inside graph is not necessary.
Answer: Figure 1 was revised according to reviewer suggestions. The title inside was removed and the units were added.
Comment /Suggestion
Lines 241 – 243 – That descriptions should be moved in section Methods due to it presents description of data analysis.
Answer: The proposed correction was made and the specific description was moved in the last sentence of 2.8 section.
Comment /Suggestion
In spite of authors stated similar correlation like e.g. Derakhshan et al. [39], in that study positive correlation was stated. It would be desirable to give better explanation of the negative correlation and of the results in table 2 which are very diverse EEqV R IC50 DPPHâ—Ź (L) and not always accordant to TPC.
Answer: With the method of IC50 of DPPH we measured the extract's concentration which caused 50% scavenging of DPPH radical. And for this reason, the lower IC50 resulted in higher extracts antioxidant activity. This is the reason that our study presented a negative correlation. Thus, our results are in line with the Derakhshan et al. [39] who present a positive correlation of the antioxidant activity with TPC.
The correlation test was performed between the mgGAE/L of the extract and the values of IC 50 of DPPH of the extracts (these results were not presented in the paper) before examination of operation costs issues presented in table 2.
Additional explanation was written in 2.5 and 3.3 sections.
Comment /Suggestion
Lines 244 – 245 – Did Derakhshan et al. [39], Mansour et 244 al. [40] and Ozgen et al. [41] use the same methods for TPC and the antioxidant activity in PPE like in present study? If no, it should be pointed out.
Answer: Following reviewers’ proposal we made the suggested additions in the lines
Comment /Suggestion
Line 250 – 266: 3.4 Modelling of the extraction of PP based on the operational costs. Mostly all that part should be moved in section Methods, as I already wrote.
Answer: We have followed the reviewer’s proposal and made the changes as already previously suggested.
Comment /Suggestion
Line 263 and in Eq.6 – EEV? or EEqV
Answer: The reviewer’s observation is correct and we changed it as EEqV in lines 210 and 211.
Comment /Suggestion
Lines 294 – 296 – Kaderides et al. [29], Xie et al. [46] and Wang et al. [47] did not use the same technique and the same raw material like in present study, so results could not be just compared without pointed out those differences.
Answer: We made the changes proposed by the reviewer, referring to the studies without making a comparison.
Comment /Suggestion
Line 321 – three question marks are written, why?
Answer: The three question marks were placed by mistake and for this reason, removed.

Reviewer 2 Report
The manuscript entitled "Optimization of vacuum microwave assisted extraction of pomegranate fruits peels and evaluation of extracts' bioactivity" presents the optimization of an extraction process carried out in an industrial scale reactor.
Although the topic is interesting and presents experiments that are not often found in the literature, my major concern is related to the way the experimental work was planned and then the experimental results explained. In this work everything is based on statistical tools: a very complex experimental design has been used (4 variables, 29 experiments). Probably a better approach should have been some screening experiments in order to find the variables that do have statistically significant effect on the responses studied, and then optimize them. In this way the experimental work could have been substantially reduced, and the experimental results better explained and understood.
Then the discussion of the experimental results is based on the ANOVA study, but not on the fundamentals of Chemical Engineering for the extraction processes, which makes the discussion of the results to be superficial and not deep enough to get useful insights that can be applied to other extraction processes carried out in industrial reactors. Then it is not very clear why the authors chose as independent variables RTPC and REEqVRIC50 and decided to optimize both parameters separately, and not find optimal conditions to maximize both simultaneously.
Author Response
Reviewer 2
Comment /Suggestion
The manuscript entitled "Optimization of vacuum microwave assisted extraction of pomegranate fruits peels and evaluation of extracts' bioactivity" presents the optimization of an extraction process carried out in an industrial scale reactor.
Although the topic is interesting and presents experiments that are not often found in the literature, my major concern is related to the way the experimental work was planned and then the experimental results explained. In this work everything is based on statistical tools: a very complex experimental design has been used (4 variables, 29 experiments). Probably a better approach should have been some screening experiments in order to find the variables that do have statistically significant effect on the responses studied, and then optimize them. In this way the experimental work could have been substantially reduced, and the experimental results better explained and understood.
Answer: The calculated response are multivariable quantities. Sometimes it is confusing that in ANOVA analysis some of the experimental factors appear not significant, But even in this case all the valuables have an effect, may be through other model components, For example if the A factor appears not significant in the ANOVA, there always is a possibility to contribute through the A x B component (interaction between a and B) or maybe A2 is significant. Therefore, we had to follow strictly the method of RSM even if we had to conduct more experiments. In RSM theory the small numbers of experimental points and the lack of repetition to get average values reduce the possibility to get reliable models.
Comment /Suggestion
Then the discussion of the experimental results is based on the ANOVA study, but not on the fundamentals of Chemical Engineering for the extraction processes, which makes the discussion of the results to be superficial and not deep enough to get useful insights that can be applied to other extraction processes carried out in industrial reactors. Then it is not very clear why the authors chose as independent variables RTPC and REEqVRIC50 and decided to optimize both parameters separately, and not find optimal conditions to maximize both simultaneously.
Answer: It is true that the paper is not emphasizing in development of novel theoretical models using the well-known Chemical Engineering principles about diffusion and mass transfer phenomena. It is actually predictive modeling but the important thing is that it provides a methodology for real industrial optimization of the extraction of PP and uses for the first-time economical functions to be optimized by high order and very accurate polynomial models and not the trivial second order RSM models which are less effective. The importance of the work is based on its applicability, without scale up, directly to the industry. As far as the optimization of the two parameters RTPC and REEqVRIC50 simultaneously, this has to be done separately because the optimization of the first ones marks the maximum extraction of polyphenols from the solid matrix without taking into account the economics of the industry, while by optimizing the second, we achieved the maximum rate of extraction, which minimizes the production cost. So, the first case is ideal in order to take a polyphenol free waste at the end of the extraction, possibly more suitable for subsequent fermentation without inhibition, and in the second case we do not desire to have the contradiction of the first and we are only seeking the maximum rate for financial reasons and not a compromise with a lower rate and lower residual PP polyphenols at the post extraction solid waste.

Round 2
Reviewer 1 Report
It is obvious that author made a big effort to improve the previous manuscript, much more than I asked for, and now is much better. Still there are some concerns. Also in answers to my comments authors included comment which was similar but not mine (The authors didn’t performed verification test from the extracts received using the optimum conditions but the triplicate experiments and the high R2 of the models can assure the high accuracy of the models.), but they did not incorporate that answer in text of manuscript on some way, what they should have do. Further, I asked authors to give short explanation why they selected DPPH method among many others. They wrote me answer, but they did not incorporate it in manuscript, what I asked. Also, authors did not write under the Table 2 that results are presented as mean ± standard deviation. At the end, it is not clear why authors Extraction rate called operational cost. The formula which they used basically derived from Qu et al. (2010) but Qu et al. did not used this term „operational cost“, just extraction rate.
Author Response
Review of the article: Optimization of vacuum microwave-assisted extraction of pomegranate fruits peels by the evaluation of extracts’ phenolic content and antioxidant activity. |
Dear editor,
the authors do acknowledge the reviewers’ comments which indeed helped us to improve the quality of our work. Please accept our answers.
Reviewer’s #1 comments
1. It is obvious that author made a big effort to improve the previous manuscript, much more than I asked for, and now is much better. Still there are some concerns. -Also in answers to my comments authors included comment which was similar but not mine (The authors didn’t performed verification test from the extracts received using the optimum conditions but the triplicate experiments and the high R2 of the models can assure the high accuracy of the models.), but they did not incorporate that answer in text of manuscript on some way, what they should have do. |
The proposed by the reviewer observation added at the Conclusions section in Lines 442-445. Thank you for the comment.
|
2. Further, I asked authors to give short explanation why they selected DPPH method among many others. They wrote me answer, but they did not incorporate it in manuscript, what I asked. |
The explanation added in Lines 150-153.
|
3. Also, authors did not write under the Table 2 that results are presented as mean ± standard deviation |
The proposed by the reviewer observation added under the Table 2.
|
4. At the end, it is not clear why authors Extraction rate called operational cost. The formula which they used basically derived from Qu et al. (2010) but Qu et al. did not used this term „operational cost“, just extraction rate. |
The extraction rate is close related with extraction’s operational costs and since we performed the study in industrial type extractor, we use this term to point it. Additionally, we added an explanation in Lines 112-113. |

Reviewer 2 Report
Although I still think it is important to study the fundamentals in chemical engineering of the presented process, the authors have indicated that the most important feature of their work is the applicability of it. Therefore, in the final version of the manuscript this purpose must be stressed in order to make it clear.
Author Response
Reviewer’s #2 comments
1. Although I still think it is important to study the fundamentals in chemical engineering of the presented process, the authors have indicated that the most important feature of their work is the applicability of it. Therefore, in the final version of the manuscript this purpose must be stressed in order to make it clear.
|
This purpose is already reported clearly in Lines 112-113 at the Introduction and also at the Conclusion section in Lines 446-450. Thank you very much for the comment. |